# SPACE-TIME ATTENTION WITH SHIFTED NON-LOCAL SEARCH

## ABSTRACT

Efficiently computing attention maps for videos is challenging due to the motion of objects between frames. While a standard non-local search is high-quality for a window surrounding each query point, the window's small size cannot accommodate motion. Methods for long-range motion use an auxiliary network to predict the most similar key coordinates as offsets from each query location. However, accurately predicting this flow field of offsets remains challenging, even for large-scale networks. Small spatial inaccuracies significantly impact the attention module's quality. This paper proposes a search strategy that combines the quality of a non-local search with the range of predicted offsets. The method, named Shifted Non-Local Search, executes a small grid search surrounding the predicted offsets to correct small spatial errors. Our method's in-place computation consumes 10 times less memory and is over 3 times faster than previous work. Experimentally, correcting the small spatial errors improves the video frame alignment quality by over 3 dB PSNR. Our search upgrades existing space-time attention modules, which improves video denoising results by 0.30 dB PSNR for a 7.5% increase in overall runtime. We integrate our space-time attention module into a UNet-like architecture to achieve state-of-the-art results on video denoising.

## 1 INTRODUCTION

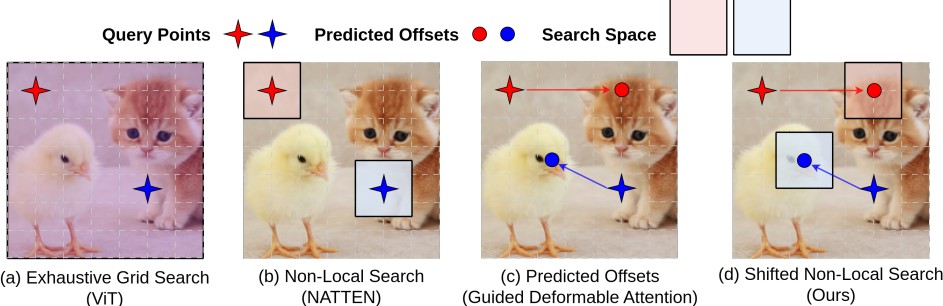

Figure 1: **Comparing the Search Space of Attention Modules.** (From left to right) ViT uses an exhaustive, global grid search which is computationally costly (Dosovitskiy et al., 2020). A non-local search can be implemented efficiently but does not shift the search space according to the motion between frames (Hassani et al., 2023). The predicted offsets used in Guided Deformable Attention allow for long-range dependencies, but the flow fields contain small spatial inaccuracies (Liang et al., 2022b). Our method, the Shifted Non-Local Search, combines the quality of a non-local search with the range of predicted offsets. It executes a small grid search surrounding the predicted offsets to correct small spatial errors.

Attention modules form data-dependent receptive fields to aggregate related features from arbitrary coordinates. This functionality is considered to be central to the success of large-scale networks (Dosovitskiy et al., 2020; Hassani et al., 2023; Tian et al., 2020; Liang et al., 2022b). Recent efforts aggregate features across frames of a video, enabling deep networks to learn temporal representations of a scene. For images, the receptive fields are often bounded by a window surrounding the query location to reduce computation and the risk of overfitting. However, across frames of a video, this window must

shift to data-dependent locations according to the motion. Long-range offsets are required, such as optical flow or nearest neighbors field (Barnes et al., 2010; Ranjan & Black, 2017a).

Non-local search strategies, such as NATTEN, provide excellent short-range receptive fields (Hassani et al., 2023). However, this category of method does not offset the search window, so it cannot handle the motion inherent to a space-time search. Alternative methods, such as Guided Deformable Attention, predict long-range offsets using an auxiliary network to accommodate motion (Liang et al., 2022b). However, accurately predicting flow fields remains an open challenge, even for large-scale networks (Butler et al., 2012).

This paper combines the quality of the non-local search with the range of predicted offsets. Our method, named Shifted Non-Local Search (Shifted-NLS), executes a small windowed grid search surrounding the predicted offset. For a marginal increase in wall-clock runtime, our search method acts as a correction step to the predicted offsets. In addition, our grid search is differentiable, which allows networks to learn long-range offsets. Our method works for attention because, unlike optical flow's goal of estimating apparent motion, standard attention modules are defined through a grid search. We show our search method improves video alignment, upgrades existing space-time attention modules, and enables a state-of-the-art architecture for video denoising.

Critically, this paper also offers a practical means to compute the Shifted Non-Local Search. An important related work, named N3Net, already offers a similar method (Plötz & Roth, 2018). However, their method is not presented in the context of attention and requires integer-spaced indexing. Also, the N3Net search's forward runtime is 3-7x slower than our search and requires over a 10-25x spike in GPU memory. These computational demands may explain why the module has not been adopted in recent works on space-time attention, and our Pytorch-friendly module offers a practical alternative (Paszke et al., 2019).

In summary, our contributions are: (i) We propose the shifted non-local search module for space-time attention. The module corrects spatial errors of predicted offsets using a high-fidelity windowed grid search. (ii) Our implementation uses in-place computation to reduce computational demands compared to previous work, using 10 times less memory and executing 3 times faster than N3Net (Plötz & Roth, 2018). While our code is not explicitly optimized for speed, our search's runtime is only 1 - 2.5 times slower than an optimized space-only non-local search (Hassani et al., 2023). (iii) Our search method improves video alignment quality by more than 3 dB PSNR, yielding improved deep network quality for video denoising.

## 2 RELATED WORKS

**Space-Only Attention:** Attention modules often use a modified search space to be computationally efficient, and most of them search only space (Dosovitskiy et al., 2020; Mou et al., 2021; Liu et al., 2021). SWIN uses an asymmetric search space but can be efficiently computed using native Pytorch code. Hassani et al. (2023) offers an efficient non-local search but cannot accommodate long-range offsets. Xia et al. (2022) applies predicted offsets for single images but suffers from the inaccuracy of using a network to predict the flow field.

**Space-Time Attention:** Recent works propose temporal attention modules using predicted offsets learned with auxiliary networks which are inspired by Deformable Convolution (Dai et al., 2017). Examples of these methods include the temporal mutual self-attention module (TSMA), the temporal deformed alignment module (TDAN), and the guided deformable attention module (GDA) (Liang et al., 2022a; Tian et al., 2020; Liang et al., 2022b). Each method predicts pixel-level offsets and warps an adjacent frame to match a query frame. These methods all require training a network to learn these offsets. Plötz & Roth (2018) proposed N3Net which does execute a shifted grid search, but its implementation is not connected to attention modules, does not propagate gradients through its grid search, and requires expensive computation. Video Non-Local Bayes is a classical method that can be formulated as an attention module (Arias & Morel, 2018). Figure 1 compares the search space of related works on a single frame.

**Restoration Architectures:** Presented concurrently with new attention modules, authors often present an architecture design for video restoration. TDAN is used for video super-resolution, and RVRT is applied to video super-resolution, deblurring, and denoising (Tian et al., 2020; Liang et al., 2022b). Their attention only applies to frame pairs, while ours searches multiple frames in parallel.

## 3 METHOD

### 3.1 PROBLEM SETUP

The attention modules described in this section introduce increasingly sophisticated search methods to establish notation and illustrate how the Shifted Non-Local Search naturally extends related works.

**Global Attention.** An input video, $\boldsymbol{X}_{\text{in}}$, has shape $T \times H \times W \times F$ denoting frames, height, width, and features. The video is projected with a $1 \times 1$ convolution to create the query ($\boldsymbol{Q}$), key ($\boldsymbol{K}$), and value ($\boldsymbol{V}$) videos. When the videos are reshaped into matrices of size $THW \times F$, we use a subscript $M$, i.e. $\boldsymbol{Q}_M$. Attention consists of two steps: search and aggregate. Searching computes the similarity between the queries and keys, often using an outer product written as the matrix $\boldsymbol{S} = \boldsymbol{Q}_M \boldsymbol{K}_M^\mathsf{T}$ with shape $THW \times THW$. Aggregation computes the weighted sum of key rows written as $\boldsymbol{A} = \sigma(\boldsymbol{S})\boldsymbol{V}_M$ with shape $THW \times F$ where $\sigma(\cdot)$ is the softmax function applied across the columns. In summary, $\boldsymbol{X}_{\text{out}} = \text{Attention}(\boldsymbol{X}_{\text{in}}) = \text{reshape}(\sigma(\boldsymbol{Q}_M \boldsymbol{K}_M^\mathsf{T})\boldsymbol{V}_M)$. The global search requires expensive computation and is unnecessary for some applications.

**Neighborhood Attention.** Neighborhood Attention constructs a sparse similarity matrix by reducing the number of similarities computed between the queries and keys (Buades et al., 2011). With specialized code, this attention is much faster than the global search and reduces the risk of overfitting. For each query, the similarity will only be computed for keys within a spatial window of size $(W_s, W_s)$ surrounding the query's coordinate. To describe this in detail, we associate the $i^{\text{th}}$ row of the similarity matrix with the 3D coordinate at $(t_i, h_i, w_i)$. The similarities are now computed as $\boldsymbol{S}[i, j] = \boldsymbol{Q}_M[i]\boldsymbol{K}_M[j]^\mathsf{T} = \boldsymbol{Q}[t_i, h_i, w_i]\boldsymbol{K}[t_j, h_j, w_j]^\mathsf{T}$ when $(t_j, h_j, w_j) \in \{(t_i, h_i - W_s/2 + \delta_h, w_i - W_s/2 + \delta_w) : \delta_h, \delta_w \in \{0, \ldots, W_s - 1\}\}$. Since most columns of $\boldsymbol{S}$ are zero, the data is restructured as $\boldsymbol{S}[i, \delta_h, \delta_w] = \boldsymbol{Q}[t_i, h_i, w_i]\boldsymbol{K}[t_i, h_i - W_s/2 + \delta_h, w_i - W_s/2 + \delta_w]^\mathsf{T}$.

**The Non-Local Search.** Rather than compute similarities between pixels, the standard non-local search from denoising literature operates on patches (Buades et al., 2011). Patches are more robust to noise than pixels and allow query coordinates to be skipped with an integer-valued query stride. The final output will be valid (e.g. no holes) when the patch size ($P$) and query stride ($S_Q$) satisfy the following condition, $\lfloor (P - 1)/2 \rfloor < S_Q$. To clean-up the messy indexing, we compactly write the spatial (height) index as $h_i(\delta_h) = h_i - W_s/2 + \delta_h$. Similarity values are now computed as $\boldsymbol{S}[i, \delta_h, \delta_w] = \sum_{p_h, p_w = -P/2, -P/2}^{P/2, P/2} \boldsymbol{Q}[t_i, h_i + p_h, w_i + p_w]\boldsymbol{K}[t_i, h_i(\delta_h + p_h), w_i(\delta_w + p_w)]^\mathsf{T}$ where $i \in \{0, \ldots, T(HW/S_Q^2) - 1\}$ so $\boldsymbol{S}$ has shape $THW/S_Q^2 \times W_s \times W_s$.

### 3.2 THE SHIFTED NON-LOCAL SEARCH

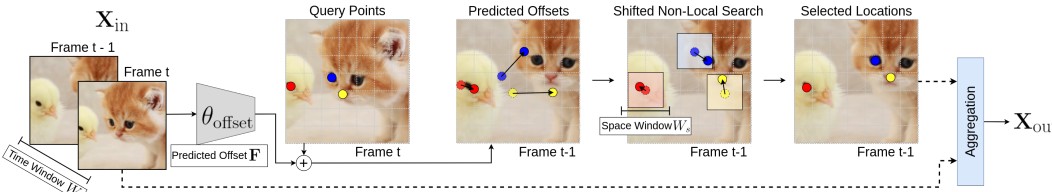

Figure 2: **The Shifted Non-Local Search for Space-Time Attention.** This figure depicts a space-time attention module using the Shifted Non-Local Search. The query points are deformed using the predicted offsets. Next, a grid search is executed surrounding the predicted offsets, and then the most similar locations are chosen from the search window. These locations are aggregated using a module such as Guided Deformable Attention.

**The Shifted Non-Local Search.** A Shifted Non-Local Search (Shifted-NLS) executes a Non-Local Search with the center of each spatial window shifted by an offset. The offsets between frames $t$ and $t - 1$ are denoted as $\boldsymbol{F}_{\text{in}}$ with shape $T \times H \times W \times 2$. The center of the search window is *shifted* from $(h_i, w_i)$ to $(h_i + \Delta_h(i), w_i + \Delta_w(i))$ with $(\Delta_h(i), \Delta_w(i)) = \boldsymbol{F}_{\text{in}}[t_i, h_i, w_i]$. This shift is depicted in Figure 2 by the colored circles at the end of the arrows under "Predicted Offsets". The similarities are computed as $\boldsymbol{S}[i, \delta_h, \delta_w] = \sum_{p_h, p_w = -P/2, -P/2}^{P/2, P/2} \boldsymbol{Q}[t_i, h_i, w_i]\boldsymbol{K}[t_i - 1, h_i(\delta_h + p_h) + \Delta_h(i), w_i(\delta_w + p_w) + \Delta_w(i)]^\mathsf{T}$ using compact notation for the spatial (height)

index, $h_i(\delta_h) = h_i - W_s/2 + \delta_h$. These offset search windows are depicted by the colored squares under "Shifted Non-Local Search" in Figure 2. The output offsets are the displacements from each query coordinate: $\boldsymbol{F}_{\text{out}}[i, \delta_h, \delta_w] = (h_i(\delta_h) + \Delta_h(i) - h_i, w_i(\delta_w) + \Delta_w(i) - w_i)$.

Once the similarities are computed, we collapse the search dimensions ($W_s \times W_s$) into a single dimension ($W_s^2$) and retain only the top-L (aka "top-K") most similar columns, $\boldsymbol{S}_L, \boldsymbol{F}_{\text{out},L} = \text{top-L}(\boldsymbol{S}, \boldsymbol{F}_{\text{out}}, L)$. The top-L operator has known theoretical issues with differentiation, but we observe networks still learn good weights despite this (Plötz & Roth, 2018). The top-L ($L = 1$) coordinates are depicted under "Selected Locations" on the far right of Figure 2. This output is written as the similarity ($\boldsymbol{S}_L$) and offset ($\boldsymbol{F}_{\text{out},L}$) tensors with shapes $T(HW)/S_Q^2 \times L$ and $T(HW)/S_Q^2 \times L \times 2$, respectively. In summary: $\boldsymbol{S}_L, \boldsymbol{F}_{\text{out},L} = \text{Shifted-NLS}(\boldsymbol{Q}, \boldsymbol{K}, \boldsymbol{F}_{\text{in}}, L)$.

In practice, the Shifted-NLS is computed in parallel across a temporal window of size $W_t$. Additionally, a key stride ($S_K$) changes the spacing between points in the grid search to allow for sub-pixel correction, $h_i(S_K\delta_h + p_h) = h_i - S_K W_s/2 + S_K\delta_h + p_h$. And since these coordinates are floating-points, bilinear interpolation is used for efficient indexing (Jeon & Kim, 2017).

**Aggregation.** The features from the Shifted-NLS are aggregated, and an example method is a weighted sum of non-local patches. The output video is initialized to zero, and each non-local patch is added in parallel (using atomic operators) weighted by a normalized similarity value. For example, writing the offsets as $(\Delta_h(i,l), \Delta_w(i,l)) = \boldsymbol{F}_{\text{out},L}[i,l]$, each patch's $(p_h, p_w)$ pixel is added as $\boldsymbol{X}_{\text{out}}[t_i, h_i + p_h, w_i + p_w] \mathrel{+}= \sum_{l=1}^{L} \sigma(\boldsymbol{S}_L)[i,l]\boldsymbol{V}[t_i - 1, h_i + \Delta_h(i,l) + p_h, w_i + \Delta_w(i,l) + p_w]$, where $\sigma(\cdot)$ is the softmax function applied across the columns. Each pixel coordinate is divided by the number of contributing terms to normalize the output. When the patch size is 1, this is logically identical to Guided Deformable Attention (GDA) (Liang et al., 2022b). And while the Shifted-NLS is compatible with GDA, GDA is limited to aggregating features from a single frame. For our Space-Time Attention Network (STAN) architecture, we would like to aggregate features across multiple frames in parallel according to learned weights, similar to PacNet (Vaksman et al., 2021). To implement this logic, we create a module to stack $L$ patches and apply 3D convolution to reduce the stack across $L$. Details are in Supplemental Section 7.

### 3.3 Why are predicted offsets not enough?

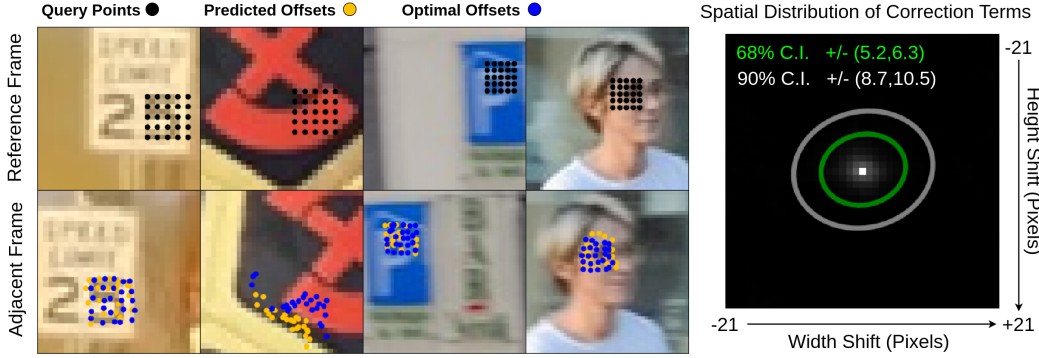

Figure 3: **Predicted offsets are only a few pixels away from their optimal location.** This figure shows query points in the query frame (top; black points), and their counterparts in the adjacent frame shifted with optical flow (bottom; blue points). The optical flow points are then corrected by a grid search of size $41 \times 41$ (bottom; yellow points). The spatial similarity between the blue and yellow points show that repurposing optical flow estimates for attention requires only small spatial corrections. The right subfigure plots the distribution of these corrections. The peak value is positioned at the center, indicating no correction is necessary for 3.5% of all cases. The two ellipses form the 68% and 90% confidence intervals.

A Shifted Non-Local Search executes a grid search surrounding a predicted offset to correct spatial inaccuracies. In this section, we explain why even this simple grid search can intuitively outperform small networks by reviewing results in the closely related research area of optical flow.

**Millions of parameters for a 6-pixel error.** The best methods for optical flow today, according to the Sintel-Clean benchmark, report an average end-point error of about 1 pixel (Butler et al.,

2012). Meanwhile, the classical pyramid-based method of 2014 reports an error of 6.73 pixels (Sun et al., 2014). Although the average improvement of about 6 pixels is impressive, this gap is closed using sophisticated training methods and network architectures with millions of parameters. Some applications claim to hugely benefit from the subpixel accuracy of these methods. However, it seems unlikely that *each instance* of an attention module will require its own auxiliary network with millions of parameters to simply predict coordinates with similar features.

**Assessing the Error of Optical Flow for Attention.** While the end-point-error is compared against an optical flow groundtruth, we qualitatively find the error to be similar when optical flow is used to estimate locations for attention. Using OpenCV's implementation of Farneback's optical flow method from 2003, Figure 3 qualitatively shows the flow's errors are concentrated in a small region surrounding the initial estimate, despite a large search grid of size $41 \times 41$ (Itseez, 2015; Farnebäck, 2003). This supports our idea to execute a small windowed grid search to correct the predicted offsets.

### 3.4 AN INPLACE COMPUTATION

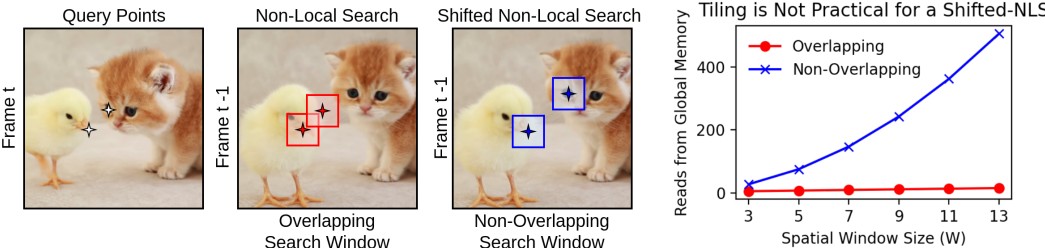

Figure 4: **Video Dynamics Challenge Existing Computational Approaches.** Searching across time is computationally challenging because *spatially adjacent patches in one frame have data-dependent spatial locations in adjacent frames*. This figure shows two neighboring locations in one frame (the chick and the kitten) move to separate spatial locations in the next frame. The benefit of NATTEN's tiling is lost because the search windows no longer overlap (Hassani et al., 2023). The rightmost subfigure plots the number of global memory reads, highlighting the lost benefit of tiling.

**Our In-Place Computation.** Our in-place computation of the Shifted Non-Local Search executes each query-key pair's similarity using the indexing from Section 3.2. The term *in-place* specifies our search does not require storing additional data related to the video. This is similar to NATTEN, but unlike N3Net which requires the construction of a patch database. However, NATTEN's fixed-window search uses tiling to reduce the number of reads from global memory, which does not freely extend to a shifted search. Also, the global memory access pattern of a shifted window search is undesirable, which necessarily increases our method's runtime. Section 4.4 shows despite this issue, our method is 3 - 7x faster than N3Net. In some cases, our search is even faster than NATTEN.

**Limitations of NATTEN.** NATTEN is designed to execute a non-local search with a small runtime (Hassani et al., 2023). Their core efficiency comes from reducing the number of reads from global memory by sharing global reads across the threads of a CUDA block. This principle does not freely extend to space-time because the search windows shift to data-dependent, non-overlapping locations, as depicted in Figure 4. Let $Q = 3$ be the tiled size and $W_s$ as the window size; then overlapping windows require only $Q + W_s - 1$ global reads while non-overlapping windows require $Q \cdot W_s^2$ global reads. The far-right subfigure in Figure 4 plots these two quantities, showing a significant disparity between the two cases. The necessarily increased number of global reads for a space-time search is a fundamental difference from space-only operators.

**Limitations of N3Net.** The query ($\boldsymbol{Q}$) and key ($\boldsymbol{K}$) videos can be *unfolded* to construct a database of patches, written as $\boldsymbol{Q}_P$ and $\boldsymbol{K}_P$ with shape $T(HW/S_Q^2) \times FP^2$ and $T(HW/S_K^2) \times FP^2$, respectively. The query ($S_Q$) and key ($S_K$) strides must be integer-valued. Normally, operators can batch across large dimensions, such as $T(HW/S_K^2)$, to control memory consumption. However, the data-dependent indexing across space-time makes batching across the keys impossible. The entire key database must be simultaneously represented in memory since each query patch may access any key patch. If queries are searched in parallel, the memory consumption increases by $P^2 \times \left(1/S_Q^2 + 1/S_K^2\right)$. For example, if $P = 3$ and $S_Q = S_K = 1$, the memory consumption of the videos increases by a factor of 18.

## 4 EXPERIMENTS

First, video alignment (Sec 4.1) demonstrates the Shifted Non-Local Search (Shifted-NLS) dramatically improves an attention module's quality. Next (Sec 4.2), RVRT's network is upgraded by replacing the Predicted Offsets with our Shifted-NLS, showing the improved attention module quality translates to improved denoising quality. Finally (Sec 4.3), RVRT's pairwise frame restriction is lifted to a multi-frame network (STAN), which achieves state-of-the-art video denoising results.

### 4.1 VIDEO FRAME ALIGNMENT

The Shifted Non-Local Search (Shifted-NLS) corrects the small spatial errors of predicted offsets (e.g. optical flow). However, assessing these spatial errors by directly comparing the offsets is misleading. Since the offsets are subsequently used for aggregation, similar offsets can (and do) produce dissimilar outputs. Video alignment provides a ground-truth target for the attention module's final output with standard qualitative and quantitative evaluation criteria.

For video alignment, we first execute the search with the queries set to frame $t$, $\boldsymbol{Q} = \boldsymbol{X}_{\text{in}}[t]$, and keys and values set to frame $t + 1$, $\boldsymbol{K} = \boldsymbol{V} = \boldsymbol{X}_{\text{in}}[t + 1]$. Second, we aggregate using only the most similar patches (top-$L = 1$). The output should match frame $t$ of the input, i.e. $\boldsymbol{X}_{\text{out}} \approx \boldsymbol{X}_{\text{in}}[t]$. This experiment uses the first 10 frames from the DAVIS training dataset (Pont-Tuset et al., 2017). When searching and computing the Farneback optical flow, we add a small amount of Gaussian noise ($\sigma^2 = 15$) to simulate the training dynamics between the query and key values (Farnebäck, 2003). Alignment quality is measured as the PSNR between the noise-free aligned and reference images. Both the Shifted-NLS and the Non-Local Search (NLS) methods use our implementation since NATTEN's patch size is fixed to 1 and limited to a search space of 13 ($W_s = 13$).

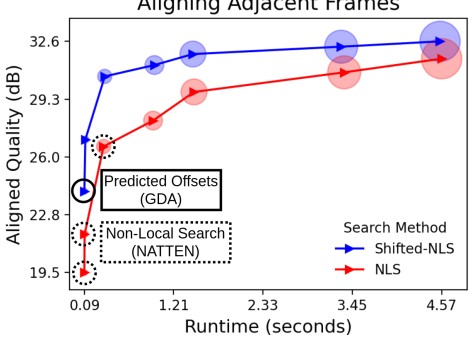

Figure 5: **The Shifted-NLS Improves Alignment Quality.** The Shifted-NLS achieves a better alignment with fewer search pixels than the NLS. Along each line the search window increases, $\{1, 3, 11, 15, 21, 27, 33\}$. The size of each point indicates the peak allocated memory when executing the search. A larger search grid is slower and consumes more memory.

Figure 6: **Video Motion Impacts the Alignment Quality.** The y-axis plots the difference in the alignment quality between the Shifted-NLS and the NLS. The x-axis is an estimate of the video's motion using the average absolute value of the optical flow. Each point represents a single video sequence. The dotted line is drawn at 0 dB PSNR: no difference.

Figure 5 compares the alignment quality and runtime of the Shifted-NLS and the NLS as the search space expands. Each point is associated with a spatial window size, $W_s \in \{1, 3, 11, 15, 21, 27, 33\}$. A window of size 1 indicates no search. Currently, NATTEN supports window sizes up to 13, as indicated by the dotted circles. For the Shifted-NLS, the PSNR plateaus around window size 11, while for the NLS it plateaus around 21. This matches our intuition that optical flow contains small spatial errors, which our grid search corrects. When the spatial search window is 11, the Shifted-NLS yields 30.60 dB PSNR while the NLS and the Predicted Offsets yield 26.63 and 24.11 dB PSNR, respectively. Figure 6 shows our Shifted-NLS method's improvement depends on video motion. Each point is the difference in PSNR between the Shifted-NLS and the NLS for each video in the DAVIS training dataset. When motion is larger than about 3 pixels, Shifted-NLS improves the alignment quality by more than 5 dB PSNR. When the average motion is less than 1 pixel, the Shifted-NLS degrades the search quality. In the case of small motion, the offset values act as noise.

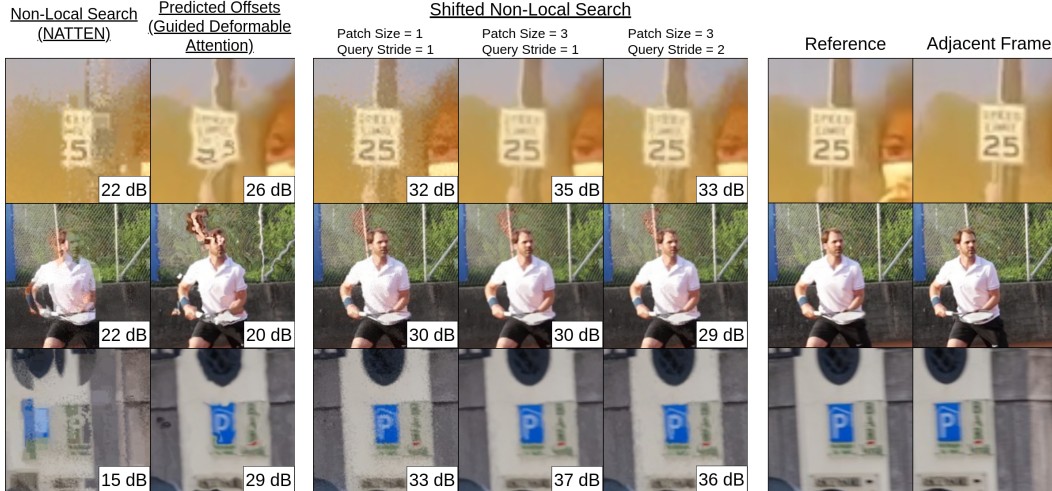

Figure 7: **Comparing alignment quality for different search methods.** This figure uses the indicated columns's search method to align the adjacent frame (right) to the reference frame (second from right). The Non-Local Search and the Shifted Non-Local Search use a search space of $11 \times 11$. The bottom-right corner reports the PSNR between the aligned and reference images; higher is better.

Figure 7 shows qualitative examples of the aligned images. For both the NLS and the Shifted-NLS methods, the spatial search window is set to $11 \times 11$. The NLS patch size is set to 1 to match NATTEN, and the Shifted-NLS patch size and query stride is indicated in the column title. The NLS method creates a doubling effect because the search radius cannot compensate for the motion shifts. For example, the first number of the speed limit sign (top row) reads "5" rather than "2". The Shifted-NLS largely removes the doubling effect, but not entirely. When the optical flow is inaccurate, a doubling effect still appears. For example, in the third row, a face appears where only a fence should be visible. The errors from Predicted Offsets create a warping effect similar to psychedelic art or the melting clocks of Salvador Dalí. The Shifted-NLS method visually removes the warping effect, replacing wavy edges with sharp ones.

Figure 7 also shows the impact of patch size and query stride. A larger patch size reduces noise since the overlapping patches are averaged together. When the query stride is 2, the pixels of each patch no longer overlap. This is qualitatively between the grainy image with patch size 1 and the smoothed image with patch size 3. When the query stride is 2, some pixels are not aligned which can reduce the overall alignment quality (middle row).

## 4.2 UPGRADING SPACE-TIME ATTENTION

This experiment shows that replacing a small neural network with our Shifted Non-Local Search improves denoising quality. Guided Deformable Attention (GDA) uses an auxiliary network to produce offsets for aggregation by transforming an input video clip and optical flow offsets: $F_{\text{out}} = $ Auxiliary Network$(X_{\text{in}}, F_{\text{in}})$. We replace their auxiliary network with our Shifted Non-Local Search: $_-, F_{\text{out},L} = $ Shifted-NLS$(X_{\text{in}}, F_{\text{in}}, L)$ with $L = 9$ to match RVRT. In this experiment, our spatial window is $9 \times 9$, the temporal window is fixed to 1 by architecture design, the query stride is 1 ($S_Q = 1$), the key stride is $1/2$ ($S_K = 1/2$), and the patch size is 1. Table 1 shows the denoising quality improves when using our search method compared to using predicted offsets. The improvement is between $0.20 - 0.40$ dB across all noise levels, an increase often attributed to a new architecture.

## 4.3 SPACE-TIME ATTENTION NETWORK (STAN)

We integrate the Shifted Non-Local Search into our Space-Time Attention Network (STAN). The architecture is a simple mixture of the UNet and RVRT networks (Ronneberger et al., 2015). We train the network for video denoising on the DAVIS train-val dataset (Pont-Tuset et al., 2017). We test the network on the DAVIS testing dataset and the Set8 dataset (Tassano et al., 2020). Due to space, we regulate details to Supplemental Section 6.4.

Table 1: **Upgrading Previous Space-Time Attention Modules.** [PSNR↑/SSIM↑/ST-RRED↓] This table reports denoising results on the DAVIS test dataset. RVRT's GDA module aggregates features according to the input offsets. This table compares using the offsets from an auxiliary network (Predicted Offsets; the default) with a small grid search (Shifted-NLS).

| $\sigma$ | Predicted Offsets | Shifted-NLS |
|----|----|----|
| 10 | 38.69/0.966/0.004 | **38.90/0.967/0.004** |
| 20 | 35.32/0.933/0.013 | **35.58/0.936/0.012** |
| 30 | 33.39/0.902/0.026 | **33.68/0.907/0.024** |
| 40 | 32.02/0.873/0.042 | **32.35/0.881/0.040** |
| 50 | 30.93/0.844/0.062 | **31.30/0.854/0.058** |
| Time (sec) | 23.86 | 25.56 |
| Mem (GB) | 10.06 | 10.06 |

Table 2 shows our network achieves state-of-the-art results on video denoising. We note the original RVRT network reports better results, but we re-train RVRT to compare both networks trained on the same number of steps. This reproducibility problem may be due to the computational environment or insufficient training time (see Supplemental Section 6.4). However, we copy RVRT's training procedure for both RVRT and STAN. Our method outperforms all other published video denoising methods, which supports the hypothesis that the Shifted Non-Local search is a useful module for space-time attention (Arias & Morel, 2018; Tassano et al., 2020; Vaksman et al., 2021).

Table 2: **State-of-the-art Video Denoising.** [PSNR↑/SSIM↑/ST-RRED↓] This table reports state-of-the-art results on video denoising. *RVRT and STAN explicitly use space-time attention. The runtime and memory usage are recorded using a single 10-frame video of resolution $480 \times 480$. We report reproduced RVRT results with further details in Supplemental Section 6.4.

|  | $\sigma^2$ | VNLB | FastDVDNet | PaCNet | RVRT (Reproduced)* | STAN* |
|----|----|----|----|----|----|----|
| | 10 | **37.26** | 36.44 | 37.06 | 36.66/0.955/0.003 | 37.19/0.960/0.002 |
| | 20 | 33.72 | 33.43 | 33.94 | 33.47/0.918/0.011 | **34.27/0.931/0.007** |
| Set8 | 30 | 31.74 | 31.68 | 32.05 | 31.65/0.885/0.022 | **32.58/0.905/0.013** |
| | 40 | 30.39 | 30.46 | 30.70 | 30.38/0.855/0.035 | **31.39/0.880/0.021** |
| | 50 | 29.24 | 29.53 | 29.66 | 29.41/0.829/0.052 | **30.46/0.856/0.030** |
| | 10 | 38.85 | 38.71 | 39.97 | 39.29/0.970/0.003 | **40.22/0.976/0.002** |
| | 20 | 35.68 | 35.77 | 36.82 | 36.00/0.942/0.010 | **37.30/0.956/0.007** |
| DAVIS | 30 | 33.73 | 34.04 | 34.79 | 34.12/0.915/0.021 | **35.54/0.937/0.012** |
| | 40 | 32.32 | 32.82 | 33.34 | 32.80/0.891/0.034 | **34.26/0.918/0.020** |
| | 50 | 31.13 | 31.86 | 32.20 | 31.78/0.868/0.050 | **33.26/0.901/0.029** |
| Time (sec) | | 497.93 | 0.11 | 182.34 | 1.63 | 3.26 |
| GPU Memory (GB) | | 0.0 | 0.37 | 12.35 | 4.25 | 10.75 |
| Parameters ($10^6$) | | N/A | 2.4 | 2.9 | 12.8 | 12.1 |

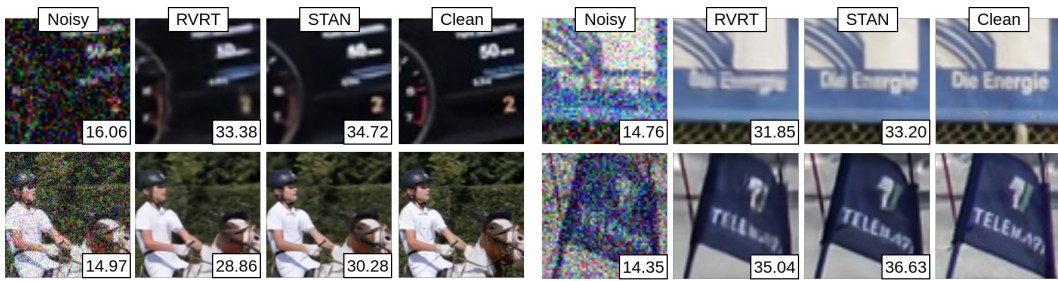

Figure 8: **Qualitatively Comparing Denoised Outputs.** [PSNR↑] RVRT and STAN use space-time attention and are trained using the same procedure. STAN recovers more small details than RVRT.

## 4.4 COMPUTATIONAL BENCHMARKING

This section compares the computation for three non-local search strategies. The Shifted-NLS and N3Net methods execute a space-time search, and NATTEN executes a space-only (unshifted) search. Each benchmark includes a function call to a `top-L` function for compatibility with existing aggregation methods. Figures 9 and 10 report benchmark results of each search method executed on

a 5 frame video with varying resolution. Due to NATTEN's tiling, its query stride is fixed at 1. The other methods vary the query stride to 1 or 2 as indicated by the dotted and solid lines, respectively.

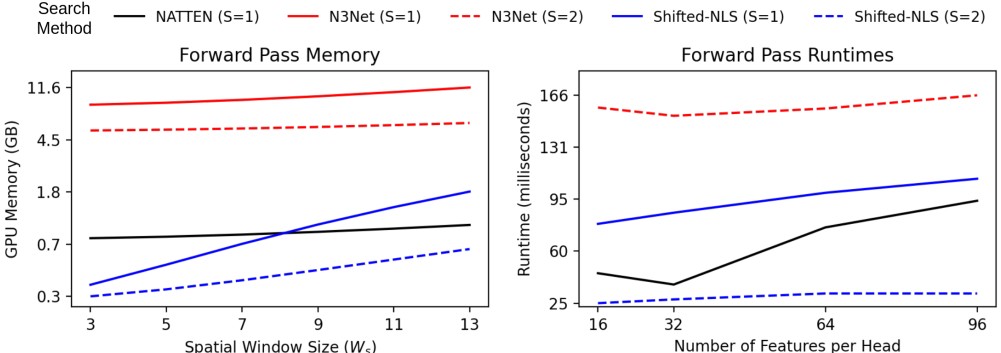

Figure 9: **The Shifted-NLS Consumes Less Memory.** Our Shifted Non-Local Search module consumes less than $20\times$ the memory of N3Net's search method and half the memory of NAT-TEN. N3Net expands the video into a database of patches, which requires an absurd consumption of GPU memory. NATTEN searches only pairs of frames, so the temporal search space must be stacked along the batch dimension.

Figure 10: **The Runtime of Shifted-NLS is Competitive.** NATTEN is a space-only search but serves as a competitive baseline. Importantly, NATTEN's efficient tiling requires a query stride of 1, unlike both other search methods. The Shifted-NLS method is $2.4$ times slower than NATTEN when the query stride is 1, but it is 2 times faster when the query stride is 2.

Figure 9 reports memory consumption for an input video with 192 features (such as in RVRT) using images with resolution $152 \times 152$. Both N3Net and Shifted-NLS use a patch size of 7. N3Net requires dramatically more memory than the Shifted-NLS module since it explicitly constructs a database of patches. When the spatial window size is 3, N3Net consumes 12.43 GB of memory, while the Shifted-NLS consumes 0.33 GB of memory. NATTEN's memory consumption grows from about $0.75$ GB to $0.97$ GB. NATTEN searches pairs of frames, so parallel searching across space-time requires stacking frames of the temporal search window along the batch dimension[1].

Figure 10 reports runtimes using images with resolution $320 \times 320$, a search window of size 9, and a patch size of $1$. As expected, the Shifted-NLS module is slower than NATTEN when the query stride is fixed to 1. For example, when the number of features is 32 the runtime of NATTEN and Shifted-NLS is about $36.77$ and $84.36$ milliseconds (ms), respectively. N3Net is far slower than both methods; N3Net's runtime for a query stride of 1 is too slow to plot clearly (about 490 ms). Notably, the Shifted-NLS is faster than NATTEN when the query stride can be set to 2. For 32 features, the runtime of the Shifted-NLS drops from $84.36$ to $27.95$ ms. However, the search quality will degrade as the query stride increases so the utility of this faster runtime depends on the application.

## 5    CONCLUSION

This paper presents a Shifted Non-Local Search module for space-time attention. We first observe the errors of offsets predicted from auxiliary networks require only small spatial corrections. Rather than train a large-scale network with millions of parameters, we propose using a small grid search to correct these errors. Our in-place implementation of the Shifted Non-Local Search avoids absurd memory spikes with a competitive runtime. Correcting the small spatial errors corresponds to over a 3 dB improvement when aligning adjacent frames. We show this translates to improved denoising quality within denoising networks. As this module is designed for learning temporal representations, future work can apply this method to additional computer vision tasks such as instance segmentation and video synthesis.

---

[1]We note this stacking of frames is not measured in NATTEN's runtime

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
