# OpenReview forum: "Space-Time Attention with Shifted Non-Local Search"
_ICLR.cc/2024/Conference — Submitted to ICLR 2024_

### Official Review · Reviewer_CKa5 · 2023-10-25

**Soundness:** 2 fair
**Presentation:** 1 poor
**Contribution:** 2 fair
**Rating:** 5
**Confidence:** 2

**Summary:**

This paper proposes a dense offset field (like optical flow) by using attention, and shows demonstrations of video frame alignment (Fig7) and video denoising (Fig8).

The paper claims to propose a kind of a combination of NATTAN (non-local search) and GDA (shift or displacement prediction) as shown in Figure 1. It first finds a large displacement somehow, then refines it using attention, as demonstrated in Fig3. Thus the proposed method is called "shifted" "non-local search".

The core of the paper is section 3.1, which shows the whole process using top-k search with attention, and section 3.2 introduces the case when shift F is given.

Section 3.3 justifies that the refinement of optical flow leads to a better offset estimation, and section 3.4 simply states that the method is implemented on CUDA (which is called in-place).

Experiments show that the proposed method gives a better quality, less memory usage (Fig9) and faster computation (Fig10).

**Strengths:**

This approach is somewhat on the hardware side and is thus very advantageous in terms of speed and memory consumption over other methods. The method is implemented "in-place", whatever it means as no details are disclosed, so fewer memory consumption is very attractive compared to recent memory-hungry large models.

Denoising performance is better than others as shown in Tables 1 and 2, and table 2 shows that the proposed method has a good trade-off between computation time and gpu memory.

**Weaknesses:**

Patch-based offset correction: as long as reading section 3.1, similarities for search are computed to each "reference locations" and "search locations", depending on strides S_Q and S_K. Given the predicted offset F, which is floating point coordinates, corrected coordinates reside in the integer grid. In experiments strides were set to 2 (probably, as it is now shown), however it is slower as shown in Table 10. There are no experiments on denoising and alignment with stride 2, it is difficult to expect the proposed method to work as better as stride 1.

Experiments: Results show that the proposed method works as expected, but so many details are not explained and hence it is hard to see how the method really works and how it behaves for different hyper-parameters under ablation studies. For example, patch size P, feature extractor for patches, details predicting offset F, K of top-K,

Insights: This paper is on "space-time attention" and top-k patches are used for the search over T frames. It should be shown that how these top-k patches are selected, how patches attend to where/when, and how corrections are improved, because such insights would be a great help for understanding the method and prompting potential following works.



Other comments:

Organization and writing: Symbols and concepts defined in section 3.1 are not well connected to the following sections and explanations, which makes the logical flow hard to understand.

Offset instead of alignment/denoising: Experimental results demonstrate how the method works in applications, however, directly investigating the learnt offset value F would be helpful for evaluating the proposed method.


Terms:

- "STAN" appears first in p5, but explained and spelled in p8.
- N_Q and N_K for Reference locations and Search locations do not match as Q is for query and K for key.

**Questions:**

see above

---

> ### Author Response · Authors · 2023-11-15
>
> - We hope the updates to Sections 3.1 and 3.2 better explain our method, and the meaning of “in-place computation”. By “in-place”, we simply mean no input video data is copied [[wiki](https://en.wikipedia.org/wiki/In-place_algorithm)]. The indexing described in Section 3.2 is directly used within our code.
>
> - We have updated the terms “reference/search” strides to “query/key” strides, respectively.
>
> - We have updated Figure 7 in Section 4.1 to address your interest regarding how parameters (such as patch size and query stride) impact the search quality. We also gently note in the supplemental Section 6.2, an experiment uses a patch size of 7 and query stride of 4. Additionally, Sections 6.3 and 6.4 include ablations experiments regarding the key stride and space-time window sizes.
>
> - Directly investigating the offsets, $\mathbf{F}$, is important. This is a motivation for Figure 3 in Section 3.3. However, directly comparing offsets to evaluate the quality of a search module will be misleading. This is because attention modules act as a kernel function on the input video. Thus different offsets may produce similar outputs, depending on the content of the value video ($\mathbf{V}$).  Written agin, when the two offsets are not equal, $\mathbf{F}^{\text{Ideal}}  \neq \mathbf{F}^{\text{Shifted-NLS}}$, the final output might be similar, $\mathbf{X}^{\text{out,Ideal}} \approx \mathbf{X}^{\text{out,Shifted-NLS}}$. It may also be true that when the two offsets are similar (but not equal), the final output might be dramatically different. This ambiguity is resolved by comparing offsets through their impact on the attention module's final output. The Video Alignment experiment in Section 4.1 provides an interpretable output target for the attention module. We thank you for pointing out this gap in our explanation and we have updated the text in Section 4.1 to clarify this point.

---

> > ### Author Response · Authors · 2023-11-22
> >
> > Dear Reviewer,
> >
> > As the discussion period nears its conclusion, we wish to ensure that all your concerns have been adequately addressed. We would greatly appreciate any additional feedback you may have. Our goal is to finalize the rebuttal with the confidence that our manuscript reflects the valuable feedback provided by the review committee.
> >
> > Thank you for your time and consideration,
> >
> > the Authors

---

### Official Review · Reviewer_UHjC · 2023-10-30

**Soundness:** 2 fair
**Presentation:** 2 fair
**Contribution:** 2 fair
**Rating:** 5
**Confidence:** 3

**Summary:**

This paper proposes Shifted Non-Local Search for frame-wise alignment. Specifically, the query points are searched in the windows which is shifted by the predicted optical flows. The top-k locations are then aggregated with Guided Deformable Attention and 3D convolution.

**Strengths:**

1. The authors demonstrate that optical flow requires only minor spatial corrections for frame-wise alignment.

2. The authors introduce In-Place Computation, which significantly reduces the memory working set and consequently enhances speed.

3. The proposed method achieves state-of-the-art results on video denosing task.

**Weaknesses:**

1. The way authors show that optical ﬂow only needs small spatial corrections is from the results of Sintel-Clean benchmark, however, this setting is far from the real-world dataset, where blur and degradation could happens. Moreover, these results are from methods with high computational cost, which is not feasible for the online setting.

2. the idea is already explored in video enhancement task, such as BasicVSR++ [1] RVRT [2], where the deformable convolutions/attentions' offsets are computed on top of the SpyNet predicted optical flow. Moreover,  IART [3] propose an cross-attention scheme by searching around the OF-shifted window.

3. This paper is the mixture of

[1] BasicVSR++: Improving Video Super-Resolution with Enhanced Propagation and Alignment (CVPR2022)

[2] Recurrent Video Restoration Transformer with Guided Deformable Attention (NeurlPS2022)

[3] An Implicit Alignment for Video Super-Resolution (arXiv:2305.00163)

**Questions:**

1. the reproduced results for RVRT 39.29 seems deviate a lot from the original paper 40.57 for DAVIS sigma=10, why this is the case? Moreover, I think RVRT already implement offsets on top of predicted optical flow, which I think is the same method with the proposed one.

2. Is there direct comparison on STAN alignment with guided deformable attention? (i.e. comparing results by only changing the alignment module, keep the original backbone unchanged for RVRT.)

---

> ### Author Response · Authors · 2023-11-15
>
> **Not a Mixture.** Our paper is not a mixture of the three listed papers.
> - First, BasicVSR++ [1] is not considered in this paper as our method is inspired from non-local denoising methods, such as Video Non-Local Bayes [4].
> - Second, RVRT [2] uses Predicted Offsets while our Shifted Non-Local Search outperforms this method by 3 - 7 dB on video alignment in Figures 6 and 7 (Sec 4.1). When replacing RVRT’s Predicted Offsets with our Shifted Non-Local Search, RVRT’s denoising quality improves by 0.30 dB in Table 1 (Sec 4.2).
> - Third, IART [3] is an unpublished paper that provides less functionality than the Shifted Non-Local Search and requires 81 times the memory ([relevant code link](https://github.com/kai422/IART/blob/05ee7d57fcc2c7d028e8389c330b50b5c797aa41/archs/implicit_alignment.py#L144)) when searching a window of size 9x9.
>
> [4] Pablo Arias and Jean-Michel Morel. Video denoising via empirical bayesian estimation of space-time patches. Journal of Mathematical Imaging and Vision, 2018.
>
> **IART.** IART [3] is an unpublished paper that is (1) computationally more expensive and (2) provides less functionality than this paper’s Shifted Non-Local Search. Firstly, IART requires a significant spike in memory consumption. When searching a spatial window of size $(W_s,W_s)$, IART must copy the input video $W_s^2$ times ([relevant code link](https://github.com/kai422/IART/blob/05ee7d57fcc2c7d028e8389c330b50b5c797aa41/archs/implicit_alignment.py#L144)). For a 9x9 search (as in Section 4.1), IART requires the input video to be copied 81 times, while the in-place Shifted Non-Local requires no copying. Second, IART’s use of Pytorch’s grid_sample function restricts the method to use image patches of size 1, while our method uses image patches of size 3 and 7 in Sections 4.1 and 6.2, respectively. In the first and third rows of Figure 7, there is an approximate 3 - 4 dB difference between a patch size of 1 and 3.
>
> **RVRT uses Predicted Offsets.** We agree that RVRT does indeed use an auxiliary network to predict offsets (named “Predicted Offset” in the paper) with optical flow and video features as inputs. In Table 1 from Section 4.2, we replace this small network with our Shifted Non-Local Search and observe the video denoising quality improve. We have updated the text in Section 4.2 to clarify this point.
>
> **Small Optical Flow Errors on the Sintel-Clean Benchmark.** We have updated our text in Section 3.3 to clarify our point regarding the Sintel-Clean dataset. We agree with you: the results from the Sintel-Clean benchmark will be overly optimistic. Our discussion in Section 3.3 is to illustrate that even recent, large-scale deep networks still report an end-point-error of about 1 even on this “far from real-world” dataset. Yet, using OpenCV’s implementation of Farnback’s 2003 optical flow method (a computationally “cheap” method) combined with a large grid search of 41x41 pixels, the most similar pixels (i.e. the pixels used for attention) exist in a small radius surrounding the offset values. We further acknowledge this radius may grow as noise increases, which is a motivation for the Video Alignment experiment in Section 4.1.
>
> **Q1. Reproducing RVRT.** One reason our reproduced RVRT network yields lower quantitative results may be due to our computational setup. We use a stand-by queue on SLURM, which requires training in 4-hour increments. While existing open-source packages claim to fully support this functionality, in reality, this may be unreliable [examples of previous issues: [1](https://github.com/Lightning-AI/lightning/issues/18588), [2](https://github.com/Lightning-AI/lightning/issues/12812)]. Still, the STAN network is trained in the same fashion and yields significantly better results. Training time may also be a factor, which we discuss in more detail in Supplement Section 6.4.
>
> **Q2. Comparing STAN and RVRT.** Since STAN processes multiple frames in parallel and RVRT does not, we are unable to directly compare the two networks. The updated text at the start of Section 4 hopefully clarifies this point. Notably, in Section 4.2 we do directly compare our proposed grid search against the auxiliary network used within the Guided Deformable Attention module (within RVRT). Table 1 shows the direct comparison between offsets computed from an auxiliary network (“Predicted Offsets” ) and our grid search (“Shifted Non-Local Search”). We also include ablation experiments for both RVRT and STAN in Sections 6.3 and 6.4, respectively.

---

> > ### Author Response · Authors · 2023-11-22
> >
> > Dear Reviewer,
> >
> > As the discussion period nears its conclusion, we wish to ensure that all your concerns have been adequately addressed. We would greatly appreciate any additional feedback you may have. Our goal is to finalize the rebuttal with the confidence that our manuscript reflects the valuable feedback provided by the review committee.
> >
> > Thank you for your time and consideration,
> >
> > the Authors

---

> ### Comment · Reviewer_UHjC · 2023-11-23
>
> I have read the responses and would like to keep my original evaluation.

---

### Official Review · Reviewer_U4AC · 2023-11-01

**Soundness:** 2 fair
**Presentation:** 2 fair
**Contribution:** 2 fair
**Rating:** 5
**Confidence:** 2

**Summary:**

The paper addresses non-local search (ie, dense point tracking) from temporal sequences. The authors propose a two stages approach, where at first a local displacement is estimated, the followed by a local search. This framework improves upon previous work which either estimate a point-wise large scale shift, or compute a a local displacement/correlation. The framework is validated in the context of space-time attention, for application such as denoising on Davis dataset.

**Strengths:**

1) Non global image matching/search is a hard problem. Applications related to video analysis (object tracking, denoising) are important.

2) The proposed approach (first predicting an off-set, then refinining the estimation in a local search window) is intuitive. The merits of the approach are shown experimentally (frame alignment, space-time attention for video denoising).

**Weaknesses:**

1) The technical explanations of the implementation of the approach are difficult to follow. Section 3.1 and 3.2 could certainly be clarified and simplified. For example,  specify the meaning of the indices, use different letters for different variables (what is the difference between I and \tilde_I?; if K_v is the variable for the Keys then do not use K again to denote the number of neighbor, etc)

2) The results section is not clear to me. I suggest the authors to start the experiments section by summarizing how they will attempt to demonstrate what are the advantages of their approach through several specific applications using different datasets and different evaluation metrics.

**Questions:**

See above

---

> ### Author Response · Authors · 2023-11-15
>
> Thank you for the comments on our presentation. We have updated Sections 3.1 and 3.2 to improve the readability of the method. In Sections 4.1 and 4.2, we have added more details to explain the intention of the experiments.
>
> - [Sections 3.1 and 3.2]: We have changed our presentation to guide the reader through increasingly sophisticated search methods within attention modules. I.e. Global Attention -> Neighborhood Attention -> Non-Local Search -> Shifted Non-Local Search.
>
> - [Section 4]: We have added a paragraph summarizing our experiment section.
>
> - [Section 4.1]: We add motivation for using the Video Alignment task as our evaluation of the different search methods.
>
> - [Section 4.2]: We add more detail to explain how RVRT’s auxiliary network (“Predicted Offsets”) are replaced with our grid search (“Shifted-NLS”).

---

> > ### Author Response · Authors · 2023-11-22
> >
> > Dear Reviewer,
> >
> > As the discussion period nears its conclusion, we wish to ensure that all your concerns have been adequately addressed. We would greatly appreciate any additional feedback you may have. Our goal is to finalize the rebuttal with the confidence that our manuscript reflects the valuable feedback provided by the review committee.
> >
> > Thank you for your time and consideration,
> >
> > the Authors

---

> > ### Comment · Reviewer_U4AC · 2023-11-23
> >
> > Thank you to the authors for the revised version of the manuscript (which as significantly improved). The paper presents a smart implementation of existing concepts combined together (cf comments of reviewer UHjC). The novelty of the current work still needs to be better exposed.

---

### Author Response · Authors · 2023-11-15

Thank you reviewers for your comments! The paper has been updated, and major updates are colored in blue. Notably, we have updated Sections 3.1 and 3.2 to improve readability. Throughout the paper, some of the text has been tightened-up to make room for the updates. Detailed changes are listed below:

- [Section 3.1,3.2] Updated the method’s presentation for clarity

- [Section 3.5] Contents are moved into Section 3.2

- [Section 4.1] Re-run Figures 5 and 6 using sigma=15  to match the text (previously sigma=0).

- [Section 4.1] Update Figure 7 to show the impact of patch size and query stride. Also, the key stride is updated from 0.5 to 1 to improve the quality of the competing, space-only Non-Local Search.

- [Section 4.2] Updated text to clarify the experiment’s goal.

- [Section 4.4] Update parameters for Figures 9 and 10 to better represent resource usage.

- [Section 6.2] Update writing in COLA-Net's experimental details and Figure 11's caption.

- [Section 6.4] Updated and corrected details on how we reproduced RVRT

- [Section 7] Update example code to be executable in native Pytorch.

---

### Meta-Review · Area_Chair_csEv · 2023-12-05

**Metareview:**

Summary: The paper proposes a shifted non-local attention for video processing, e.g. video denoising, frame alignment.
Strength: The approach is intuitive (estimate an offset, then refine within an offset-ed local window), which is shown to be advantageous in memory (10x lower) and speed (3x faster). The method also shows better performance than competing methods.
Weakness: Technical explanations are a little convoluted and can be further simplified. It’s true the authors made a major effort in updating the methodology section. I found the work still a bit convoluted to follow. At the high-level its objectives and proposals are clear, the details are a bit hard for me to follow. Concerns also remain on novelty, for instance I agree with reviewer UHiC who pointed out that prior work exists that explores this idea, e.g. deformable convolutions + SpyNet. The rebuttal has not convinced me to overwrite the reviewer's concerns.

**Justification For Why Not Higher Score:**

The paper has weaknesses that need to be addressed. The reviewers have read the rebuttal and maintained their scores. The paper received 3x below acceptance threshold. I think the paper has interesting contributions, but may not be above the acceptance threshold for ICLR. I encourage authors to read the reviewers comments, and update the paper, and submit in future venues.

**Justification For Why Not Lower Score:**

N/A

---

### Decision · Program_Chairs · 2024-01-16

Reject